# Fast Algorithms for Gaussian Noise Invariant Independent Component Analysis

**James Voss**
Ohio State University
Computer Science and Engineering,
2015 Neil Avenue, Dreese Labs 586.
Columbus, OH 43210
*vossj@cse.ohio-state.edu*

Luis Rademacher
Ohio State University
Computer Science and Engineering,
2015 Neil Avenue, Dreese Labs 495.
Columbus, OH 43210
*lrademac@cse.ohio-state.edu*

Mikhail Belkin
Ohio State University
Computer Science and Engineering,
2015 Neil Avenue, Dreese Labs 597.
Columbus, OH 43210
*mbelkin@cse.ohio-state.edu*

## Abstract

The performance of standard algorithms for Independent Component Analysis quickly deteriorates under the addition of Gaussian noise. This is partially due to a common first step that typically consists of whitening, i.e., applying Principal Component Analysis (PCA) and rescaling the components to have identity covariance, which is not invariant under Gaussian noise.

In our paper we develop the first practical algorithm for Independent Component Analysis that is provably invariant under Gaussian noise. The two main contributions of this work are as follows:
1. We develop and implement an efficient, Gaussian noise invariant decorrelation (quasi-orthogonalization) algorithm using Hessians of the cumulant functions.
2. We propose a very simple and efficient fixed-point GI-ICA (Gradient Iteration ICA) algorithm, which is compatible with quasi-orthogonalization, as well as with the usual PCA-based whitening in the noiseless case. The algorithm is based on a special form of gradient iteration (different from gradient descent). We provide an analysis of our algorithm demonstrating fast convergence following from the basic properties of cumulants. We also present a number of experimental comparisons with the existing methods, showing superior results on noisy data and very competitive performance in the noiseless case.

## 1 Introduction and Related Works

In the Blind Signal Separation setting, it is assumed that observed data is drawn from an unknown distribution. The goal is to recover the latent signals under some appropriate structural assumption. A prototypical setting is the so-called *cocktail party problem*: in a room, there are $d$ people speaking simultaneously and $d$ microphones, with each microphone capturing a superposition of the voices. The objective is to recover the speech of each individual speaker. The simplest modeling assumption is to consider each speaker as producing a signal that is a random variable independent of the others, and to take the superposition to be a linear transformation independent of time. This leads to the following formalization: We observe samples from a random vector $\mathbf{x}$ distributed according to the equation $\mathbf{x} = A\mathbf{s} + \mathbf{b} + \boldsymbol{\eta}$ where $A$ is a linear mixing matrix, $\mathbf{b} \in \mathbb{R}^d$ is a constant vector, $\mathbf{s}$ is a latent random vector with independent coordinates, and $\boldsymbol{\eta}$ is an unknown random noise independent

of **s**. For simplicity, we assume $A \in \mathbb{R}^{d \times d}$ is square and of full rank. The latent components of **s** are viewed as containing the information describing the makeup of the observed signal (voices of individual speakers in the cocktail party setting). The goal of Independent Component Analysis is to approximate the matrix $A$ in order to recover the latent signal **s**. In practice, most methods ignore the noise term, leaving the simpler problem of recovering the mixing matrix $A$ when $\mathbf{x} = A\mathbf{s}$ is observed.

Arguably the two most widely used ICA algorithms are FastICA [13] and JADE [6]. Both of these algorithms are based on a two step process:
(1) The data is centered and *whitened*, that is, made to have identity covariance matrix. This is typically done using principal component analysis (PCA) and rescaling the appropriate components. In the noiseless case this procedure orthogonalizes and rescales the independent components and thus recovers $A$ up to an unknown orthogonal matrix $R$.
(2) Recover the orthogonal matrix $R$.

Most practical ICA algorithms differ only in the second step. In FastICA, various objective functions are used to perform a projection pursuit style algorithm which recovers the columns of $R$ one at a time. JADE uses a fourth-cumulant based technique to simultaneously recover all columns of $R$.

Step 1 of ICA is affected by the addition of a Gaussian noise. Even if the noise is white (has a scalar times identity covariance matrix) the PCA-based whitening procedure can no longer guarantee the whitening of the underlying independent components. Hence, the second step of the process is no longer justified. This failure may be even more significant if the noise is not white, which is likely to be the case in many practical situations. Recent theoretical developments (see, [2] and [3]) consider the case where the noise $\boldsymbol{\eta}$ is an arbitrary (not necessarily white) additive Gaussian variable drawn independently from **s**.

In [2], it was observed that certain cumulant-based techniques for ICA can still be applied for the second step if the underlying signals can be orthogonalized.[1] Orthogonalization of the latent signals (quasi-orthogonalization) is a significantly less restrictive condition as it does not force the underlying signal to have identity covariance (as in whitening in the noiseless case). In the noisy setting, the usual PCA cannot achieve quasi-orthogonalization as it will whiten the mixed signal, but not the underlying components. In [3], we show how quasi-orthogonalization can be achieved in a noise-invariant way through a method based on the fourth-order cumulant tensor. However, a direct implementation of that method requires estimating the full fourth-order cumulant tensor, which is computationally challenging even in relatively low dimensions. In this paper we derive a practical version of that algorithm based on directional Hessians of the fourth univariate cumulant, thus reducing the complexity dependence on the data dimensionality from $d^4$ to $d^3$, and also allowing for a fully vectorized implementation.

We also develop a fast and very simple gradient iteration (not to be confused with gradient descent) algorithm, GI-ICA, which is compatible with the quasi-orthogonalization step and can be shown to have convergence of order $r - 1$, when implemented using a univariate cumulant of order $r$. For the cumulant of order four, commonly used in practical applications, we obtain cubic convergence. We show how these convergence rates follow directly from the properties of the cumulants, which sheds some light on the somewhat surprising cubic convergence seen in fourth-order based ICA methods [13, 18, 22]. The update step has complexity $O(Nd)$ where $N$ is the number of samples, giving a total algorithmic complexity of $O(Nd^3)$ for step 1 and $O(Nd^2t)$ for step 2, where $t$ is the number of iterations for convergence in the gradient iteration.

Interestingly, while the techniques are quite different, our gradient iteration algorithm turns out to be closely related to Fast ICA in the noiseless setting, in the case when the data is whitened and the cumulants of order three or four are used. Thus, GI-ICA can be viewed as a generalization (and a conceptual simplification) of Fast ICA for more general quasi-orthogonalized data.

We present experimental results showing superior performance in the case of data contaminated by Gaussian noise and very competitive performance for clean data. We also note that the GI-ICA algorithms are fast in practice, allowing us to process (decorrelate and detect the independent

components) $100\,000$ points in dimension 5 in well under a second on a standard desktop computer. Our Matlab implementation of GI-ICA is available for download at `http://sourceforge.net/projects/giica/`.

Finally, we observe that our method is partially compatible with the robust cumulants introduced in [20]. We briefly discuss how GI-ICA can be extended using these noise-robust techniques for ICA to reduce the impact of sparse noise.

The paper is organized as follows. In section 2, we discuss the relevant properties of cumulants, and discuss results from prior work which allows for the quasi-orthogonalization of signals with non-zero fourth cumulant. In section 3, we discuss the connection between the fourth-order cumulant tensor method for quasi-orthogonalization discussed in section 2 with Hessian-based techniques seen in [2] and [11]. We use this connection to create a more computationally efficient and practically implementable version of the quasi-orthogonalization algorithm discussed in section 2. In section 4, we discuss new, fast, projection-pursuit style algorithms for the second step of ICA which are compatible with quasi-orthogonalization. In order to simplify the presentation, all algorithms are stated in an abstract form as if we have exact knowledge of required distribution parameters. Section 5 discusses the estimators of required distribution parameters to be used in practice. Section 6 discusses numerical experiments demonstrating the applicability of our techniques.

**Related Work.** The name Independent Component Analysis refers to a broad range of algorithms addressing the blind signal separation problem as well as its variants and extensions. There is an extensive literature on ICA in the signal processing and machine learning communities due to its applicability to a variety of important practical situations. For a comprehensive introduction see the books [8, 14]. In this paper we develop techniques for dealing with noisy data by introducing new and more efficient techniques for quasi-orthogonalization and subsequent component recovery. The quasi-orthogonalization step was introduced in [2], where the authors proposed an algorithm for the case when the fourth cumulants of all independent components are of the same sign. A general algorithm with complete theoretical analysis was provided in [3]. That algorithm required estimating the full fourth-order cumulant tensor.

We note that Hessian based techniques for ICA were used in [21, 2, 11], with [11] and [2] using the Hessian of the fourth-order cumulant. The papers [21] and [11] proposed interesting randomized one step noise-robust ICA algorithms based on the cumulant generating function and the fourth cumulant respectively in primarily theoretical settings. The gradient iteration algorithm proposed is closely related to the work [18], which provides a gradient-based algorithm derived from the fourth moment with cubic convergence to learn an unknown parallelepiped in a cryptographic setting. For the special case of the fourth cumulant, the idea of gradient iteration has appeared in the context of FastICA with a different justification, see e.g. [16, Equation 11 and Theorem 2]. We also note the work [12], which develops methods for Gaussian noise-invariant ICA under the assumption that the noise parameters are known. Finally, there are several papers that considered the problem of performing PCA in a noisy framework. [5] gives a provably robust algorithm for PCA under a sparse noise model. [4] performs PCA robust to white Gaussian noise, and [9] performs PCA robust to white Gaussian noise and sparse noise.

## 2   Using Cumulants to Orthogonalize the Independent Components

**Properties of Cumulants:** Cumulants are similar to moments and can be expressed in terms of certain polynomials of the moments. However, cumulants have additional properties which allow independent random variables to be algebraically separated. We will be interested in the fourth order multi-variate cumulants, and univariate cumulants of arbitrary order. Denote by $Q_{\mathbf{x}}$ the fourth order cumulant tensor for the random vector $\mathbf{x}$. So, $(Q_{\mathbf{x}})_{ijkl}$ is the cross-cumulant between the random variables $x_i, x_j, x_k$, and $x_l$, which we alternatively denote as $\mathrm{Cum}(x_i, x_j, x_k, x_l)$. Cumulant tensors are symmetric, i.e. $(Q_{\mathbf{x}})_{ijkl}$ is invariant under permutations of indices. Multivariate cumulants have the following properties (written in the case of fourth order cumulants):

1. (Multilinearity) $\mathrm{Cum}(\alpha x_i, x_j, x_k, x_l) = \alpha\,\mathrm{Cum}(x_i, x_j, x_k, x_l)$ for random vector $\mathbf{x}$ and scalar $\alpha$. If $y$ is a random variable, then $\mathrm{Cum}(x_i + y, x_j, x_k, x_l) = \mathrm{Cum}(x_i, x_j, x_k, x_l) + \mathrm{Cum}(y, x_j, x_k, x_l)$.
2. (Independence) If $x_i$ and $x_j$ are independent random variables, then $\mathrm{Cum}(x_i, x_j, x_k, x_l) = 0$. When $\mathbf{x}$ and $\mathbf{y}$ are independent, $Q_{\mathbf{x}+\mathbf{y}} = Q_{\mathbf{x}} + Q_{\mathbf{y}}$.
3. (Vanishing Gaussian) Cumulants of order 3 and above are zero for Gaussian random variables.

The first order cumulant is the mean, and the second order multivariate cumulant is the covariance matrix. We will denote by $\kappa_r(x)$ the order-$r$ univariate cumulant, which is equivalent to the cross-cumulant of $x$ with itself $r$ times: $\kappa_r(x) := \mathrm{Cum}(x, x, \ldots, x)$ (where $x$ appears $r$ times). Univariate $r$-cumulants are additive for independent random variables, i.e. $\kappa_r(x + y) = \kappa_r(x) + \kappa_r(y)$, and homogeneous of degree $r$, i.e. $\kappa_r(\alpha x) = \alpha^r \kappa_r(x)$.

**Quasi-Orthogonalization Using Cumulant Tensors.** Recalling our original notation, $\mathbf{x} = A\mathbf{s} + \mathbf{b} + \boldsymbol{\eta}$ gives the generative ICA model. We define an operation of fourth-order tensors on matrices: For $Q \in \mathbb{R}^{d \times d \times d \times d}$ and $M \in \mathbb{R}^{d \times d}$, $Q(M)$ is the matrix such that

$$Q(M)_{ij} := \sum_{k=1}^{d} \sum_{l=1}^{d} Q_{ijkl} m_{lk} . \tag{1}$$

We can use this operation to orthogonalize the latent random signals.

**Definition 2.1.** A matrix $W$ is called a *quasi-orthogonalization* matrix if there exists an orthogonal matrix $R$ and a nonsingular diagonal matrix $D$ such that $WA = RD$.

We will need the following results from [3]. Here we use $A_q$ to denote the $q^{\text{th}}$ column of $A$.

**Lemma 2.2.** *Let $M \in \mathbb{R}^{d \times d}$ be an arbitrary matrix. Then, $Q_{\mathbf{x}}(M) = ADA^T$ where $D$ is a diagonal matrix with entries $d_{qq} = \kappa_4(s_q) A_q^T M A_q$.*

**Theorem 2.3.** *Suppose that each component of $\mathbf{s}$ has non-zero fourth cumulant. Let $M = Q_{\mathbf{x}}(I)$, and let $C = Q_{\mathbf{x}}(M^{-1})$. Then $C = ADA^T$ where $D$ is a diagonal matrix with entries $d_{qq} = 1/\|A_q\|_2^2$. In particular, $C$ is positive definite, and for any factorization $BB^T$ of $C$, $B^{-1}$ is a quasi-orthogonalization matrix.*

## 3 Quasi-Orthogonalization using Cumulant Hessians

We have seen in Theorem 2.3 a tensor-based method which can be used to quasi-orthogonalize observed data. However, this method naïvely requires the estimation of $O(d^4)$ terms from data. There is a connection between the cumulant Hessian-based techniques used in ICA [2, 11] and the tensor-based technique for quasi-orthogonalization described in Theorem 2.3 that allows the tensor-method to be rewritten using a series of Hessian operations. We make this connection precise below. The Hessian version requires only $O(d^3)$ terms to be estimated from data and simplifies the computation to consist of matrix and vector operations.

Let $\mathcal{H}_{\mathbf{u}}$ denote the Hessian operator with respect to a vector $\mathbf{u} \in \mathbb{R}^d$. The following lemma connects Hessian methods with our tensor-matrix operation (a special case is discussed in [2, Section 2.1]).

**Lemma 3.1.** $\mathcal{H}_{\mathbf{u}}(\kappa_4(\mathbf{u}^T \mathbf{x})) = ADA^T$ *where* $d_{qq} = 12(\mathbf{u}^T A_q)^2 \kappa_4(s_q)$.

In Lemma 3.1, the diagonal entries can be rewritten as $d_{qq} = 12\kappa_4(s_q)(A_q^T(\mathbf{u}\mathbf{u}^T)A_q)$. By comparing with Lemma 2.2, we see that applying $Q_{\mathbf{x}}$ against a symmetric, rank one matrix $\mathbf{u}\mathbf{u}^T$ can be rewritten in terms of the Hessian operations: $Q_{\mathbf{x}}(\mathbf{u}\mathbf{u}^T) = \frac{1}{12}\mathcal{H}_{\mathbf{u}}(\kappa_4(\mathbf{u}^T \mathbf{x}))$. This formula extends to arbitrary symmetric matrices by the following Lemma.

**Lemma 3.2.** *Let $M$ be a symmetric matrix with eigen decomposition $U\Lambda U^T$ such that $U = (\mathbf{u}_1, \mathbf{u}_2, \ldots, \mathbf{u}_d)$ and $\Lambda = \mathrm{diag}(\lambda_1, \lambda_2, \ldots, \lambda_d)$. Then, $Q_{\mathbf{x}}(M) = \frac{1}{12} \sum_{i=1}^{d} \lambda_i \mathcal{H}_{\mathbf{u}_i} \kappa_4(\mathbf{u}_i^T \mathbf{x})$.*

The matrices $I$ and $M^{-1}$ in Theorem 2.3 are symmetric. As such, the tensor-based method for quasi-orthogonalization can be rewritten using Hessian operations. This is done in Algorithm 1.

## 4 Gradient Iteration ICA

In the preceding sections, we discussed techniques to quasi-orthogonalize data. For this section, we will assume that quasi-orthogonalization is accomplished, and discuss deflationary approaches that can quickly recover the directions of the independent components. Let $W$ be a quasi-orthogonalization matrix. Then, define $\mathbf{y} := W\mathbf{x} = WA\mathbf{s} + W\boldsymbol{\eta}$. Note that since $\boldsymbol{\eta}$ is Gaussian noise, so is $W\boldsymbol{\eta}$. There exists a rotation matrix $R$ and a diagonal matrix $D$ such that $WA = RD$. Let $\tilde{\mathbf{s}} := D\mathbf{s}$. The coordinates of $\tilde{\mathbf{s}}$ are still independent random variables. Gaussian noise makes recovering the scaling matrix $D$ impossible. We aim to recover the rotation matrix $R$.

---

**Algorithm 1** Hessian-based algorithm to generate a quasi-orthogonalization matrix.

---

1: **function** FINDQUASIORTHOGONALIZATIONMATRIX($\mathbf{x}$)
2:     Let $M = \frac{1}{12} \sum_{i=1}^{d} \mathcal{H}_{\mathbf{u}} \kappa_4(\mathbf{u}^T \mathbf{x})|_{\mathbf{u}=\mathbf{e}_i}$. See Equation (4) for the estimator.
3:     Let $U\Lambda U^T$ give the eigendecomposition of $M^{-1}$
4:     Let $C = \sum_{i=1}^{d} \lambda_i \mathcal{H}_{\mathbf{u}} \kappa_4(\mathbf{u}^T \mathbf{x})|_{\mathbf{u}=U_i}$. See Equation (4) for the estimator.
5:     Factorize $C$ as $BB^T$.
6:     return $B^{-1}$
7: **end function**

---

To see why recovery of $D$ is impossible, we note that a white Gaussian random variable $\boldsymbol{\eta}_1$ has independent components. It is impossible to distinguish between the case where $\boldsymbol{\eta}_1$ is part of the signal, i.e. $WA(\mathbf{s} + \boldsymbol{\eta}_1) + W\boldsymbol{\eta}$, and the case where $A\boldsymbol{\eta}_1$ is part of the additive Gaussian noise, i.e. $WA\mathbf{s} + W(A\boldsymbol{\eta}_1 + \boldsymbol{\eta})$, when $\mathbf{s}$, $\boldsymbol{\eta}_1$, and $\boldsymbol{\eta}$ are drawn independently. In the noise-free ICA setting, the latent signal is typically assumed to have identity covariance, placing the scaling information in the columns of $A$. The presence of additive Gaussian noise makes recovery of the scaling information impossible since the latent signals become ill-defined. Following the idea popularized in FastICA, we will discuss a deflationary technique to recover the columns of $R$ one at a time.

**Fast Recovery of a Single Independent Component.** In the deflationary approach, a function $f$ is fixed that acts upon a directional vector $\mathbf{u} \in \mathbb{R}^d$. Based on some criterion (typically maximization or minimization of $f$), an iterative optimization step is performed until convergence. This technique was popularized in FastICA, which is considered fast for the following reasons:
1. As an approximate Newton method, FastICA requires computation of $\nabla_{\mathbf{u}} f$ and a quick-to-compute estimate of $(\mathcal{H}_{\mathbf{u}}(f))^{-1}$ at each iterative step. Due to the estimate, the computation runs in $O(Nd)$ time, where $N$ is the number of samples.
2. The iterative step in FastICA has local quadratic order convergence using arbitrary functions, and global cubic-order convergence when using the fourth cumulant [13].

We note that cubic convergence rates are not unique to FastICA and have been seen using gradient descent (with the correct step-size) when choosing $f$ as the fourth moment [18]. Our proposed deflationary algorithm will be comparable with FastICA in terms of computational complexity, and the iterative step will take on a conceptually simpler form as it only relies on $\nabla_{\mathbf{u}} \kappa_r$. We provide a derivation of fast convergence rates that relies entirely on the properties of cumulants. As cumulants are invariant with respect to the additive Gaussian noise, the proposed methods will be admissible for both standard and noisy ICA.

While cumulants are essentially unique with the additivity and homogeneity properties [17] when no restrictions are made on the probability space, the preprocessing step of ICA gives additional structure (like orthogonality and centering), providing additional admissible functions. In particular, [20] designs "robust cumulants" which are only minimally effected by sparse noise. Welling's robust cumulants have versions of the additivity and homogeneity properties, and are consistent with our update step. For this reason, we will state our results in greater generality.

Let $G$ be a function of univariate random variables that satisfies the additivity, degree-$r$ ($r \geq 3$) homogeneity, and (for the noisy case) the vanishing Gaussians properties of cumulants. Then for a generic choice of input vector $\mathbf{v}$, Algorithm 2 will demonstrate order $r-1$ convergence. In particular, if $G$ is $\kappa_3$, then we obtain quadratic convergence; and if $G$ is $\kappa_4$, we obtain cubic convergence. Lemma 4.1 helps explain why this is true.

**Lemma 4.1.** $\nabla_{\mathbf{v}} G(\mathbf{v} \cdot \mathbf{y}) = r \sum_{i=1}^{d} (\mathbf{v} \cdot R_i)^{r-1} G(\tilde{s}_i) R_i$.

If we consider what is happening in the basis of the columns of $R$, then up to some multiplicative constant, each coordinate is raised to the $r-1$ power and then renormalized during each step of Algorithm 2. This ultimately leads to the order $r-1$ convergence.

**Theorem 4.2.** *If for a unit vector input $\mathbf{v}$ to Algorithm 2 $h = \arg\max_i |(\mathbf{v} \cdot R_i)^{r-2} G(\tilde{s}_i)|$ has a unique answer, then $\mathbf{v}$ has order $r-1$ convergence to $R_h$ up to sign. In particular, if the following conditions are met: (1) There exists a coordinate random variable $s_i$ of $\mathbf{s}$ such that $G(s_i) \neq 0$. (2) $\mathbf{v}$ inputted into Algorithm 2 is chosen uniformly at random from the unit sphere $S^{d-1}$. Then Algorithm 2 converges to a column of $R$ (up to sign) almost surely, and convergence is of order $r-1$.*

**Algorithm 2** A fast algorithm to recover a single column of $R$ when $\mathbf{v}$ is drawn generically from the unit sphere. Equations (2) and (3) provide $k$-statistic based estimates of $\nabla_\mathbf{v}\kappa_3$ and $\nabla_\mathbf{v}\kappa_4$, which can be used as practical choices of $\nabla_\mathbf{v}G$ on real data.

---

1: **function** GI-ICA($\mathbf{v}, \mathbf{y}$)
2:     **repeat**
3:         $\mathbf{v} \leftarrow \nabla_\mathbf{v}G(\mathbf{v}^T\mathbf{y})$
4:         $\mathbf{v} \leftarrow \mathbf{v}/\|\mathbf{v}\|_2$
5:     **until** Convergence **return** $\mathbf{v}$
6: **end function**

---

**Algorithm 3** Algorithm for ICA in the presence of Gaussian noise. $\tilde{A}$ recovers $A$ up to column order and scaling. $R^TW$ is the demixing matrix for the observed random vector $\mathbf{x}$.

---

**function** GAUSSIANROBUSTICA($G, \mathbf{x}$)
    $W = $ FINDQUASIORTHOGONALIZATIONMATRIX($\mathbf{x}$)
    $\mathbf{y} = W\mathbf{x}$
    $R\_columns = \emptyset$
    **for** $i = 1$ to $d$ **do**
        Draw $\mathbf{v}$ from $S^{d-1} \cap \text{span}(R\_columns)^\perp$ uniformly at random.
        $R\_columns = R\_columns \cup \{\text{GI-ICA}(\mathbf{v}, \mathbf{y})\}$
    **end for**
    Construct a matrix $R$ using the elements of $R\_columns$ as columns.
    $\tilde{\mathbf{s}} = R^T\mathbf{y}$
    $\tilde{A} = (R^TW)^{-1}$
    **return** $\tilde{A}, \tilde{\mathbf{s}}$
**end function**

---

By convergence up to sign, we include the possibility that $\mathbf{v}$ oscillates between $R_h$ and $-R_h$ on alternating steps. This can occur if $G(\tilde{s}_i) < 0$ and $r$ is odd. Due to space limitations, the proof is omitted.

**Recovering all Independent Components.** As a Corollary to Theorem 4.2 we get:

**Corollary 4.3.** *Suppose $R_1, R_2, \ldots, R_k$ are known for some $k < d$. Suppose there exists $i > k$ such that $G(s_i) \neq 0$. If $\mathbf{v}$ is drawn uniformly at random from $S^{d-1} \cap \text{span}(R_1, \ldots, R_k)^\perp$ where $S^{d-1}$ denotes the unit sphere in $\mathbb{R}^d$, then Algorithm 2 with input $\mathbf{v}$ converges to a new column of $R$ almost surely.*

Since the indexing of $R$ is arbitrary, Corollary 4.3 gives a solution to noisy ICA, in Algorithm 3. In practice (not required by the theory), it may be better to enforce orthogonality between the columns of $R$, by orthogonalizing $\mathbf{v}$ against previously found columns of $R$ at the end of each step in Algorithm 2. We expect the fourth or third cumulant function will typically be chosen for $G$.

## 5 Time Complexity Analysis and Estimation of Cumulants

To implement Algorithms 1 and 2 requires the estimation of functions from data. We will limit our discussion to estimation of the third and fourth cumulants, as lower order cumulants are more statistically stable to estimate than higher order cumulants. $\kappa_3$ is useful in Algorithm 2 for non-symmetric distributions. However, since $\kappa_3(s_i) = 0$ whenever $s_i$ is a symmetric distribution, it is plausible that $\kappa_3$ would not recover all columns of $R$. When $\mathbf{s}$ is suspected of being symmetric, it is prudent to use $\kappa_4$ for $G$. Alternatively, one can fall back to $\kappa_4$ from $\kappa_3$ when $\kappa_3$ is detected to be near 0.

Denote by $z^{(1)}, z^{(2)}, \ldots, z^{(N)}$ the observed samples of a random variable $z$. Given a sample, each cumulant can be estimated in an unbiased fashion by its $k$-statistic. Denote by $k_r(z^{(i)})$ the $k$-statistic sample estimate of $\kappa_r(z)$. Letting $m_r(z^{(i)}) := \frac{1}{N}\sum_{i=1}^{N}(z^{(i)} - \bar{z})^r$ give the $r^{\text{th}}$ sample central moment, then

$$k_3(z^{(i)}) := \frac{N^2 m_3(z^{(i)})}{(N-1)(N-2)} \, , \quad k_4(z^{(i)}) := N^2 \frac{(N+1)m_4(z^{(i)}) - 3(N-1)m_2(z^{(i)})^2}{(N-1)(N-2)(N-3)}$$

gives the third and fourth $k$-statistics [15]. However, we are interested in estimating the gradients (for Algorithm 2) and Hessians (for Algorithm 1) of the cumulants rather than the cumulants themselves. The following Lemma shows how to obtain unbiased estimates:

**Lemma 5.1.** *Let $\mathbf{z}$ be a $d$-dimensional random vector with finite moments up to order $r$. Let $\mathbf{z}^{(i)}$ be an iid sample of $\mathbf{z}$. Let $\alpha \in \mathbb{N}^d$ be a multi-index. Then $\partial_{\mathbf{u}}^\alpha k_r(\mathbf{u} \cdot \mathbf{z}^{(i)})$ is an unbiased estimate for $\partial_{\mathbf{u}}^\alpha \kappa_r(\mathbf{u} \cdot \mathbf{z})$.*

If we mean-subtract (via the sample mean) all observed random variables, then the resulting estimates are:

$$\nabla_{\mathbf{u}} k_3(\mathbf{u} \cdot \mathbf{y}) = (N-1)^{-1}(N-2)^{-1} 3N \sum_{i=1}^N (\mathbf{u} \cdot \mathbf{y}^{(i)})^2 \mathbf{y}^{(i)} \tag{2}$$

$$\nabla_{\mathbf{u}} k_4(\mathbf{u} \cdot \mathbf{y}) = \frac{N^2}{(N-1)(N-2)(N-3)} \left\{ 4\frac{N+1}{N} \left( \sum_{i=1}^N ((\mathbf{u} \cdot \mathbf{y}^{(i)}))^3 \mathbf{y}^{(i)} \right) \right.$$
$$\left. -12\frac{N-1}{N^2} \left( \sum_{i=1}^N (\mathbf{u} \cdot \mathbf{y}^{(i)})^2 \right) \left( \sum_{i=1}^N (\mathbf{u} \cdot \mathbf{y}^{(i)}) \mathbf{y}^{(i)} \right) \right\} \tag{3}$$

$$\mathcal{H}_{\mathbf{u}} k_4(\mathbf{u} \cdot \mathbf{x}) = \frac{12N^2}{(N-1)(N-2)(N-3)} \left\{ \frac{N+1}{N} \sum_{i=1}^N ((\mathbf{u} \cdot \mathbf{x}^{(i)}))^2 (\mathbf{x}\mathbf{x}^T)^{(i)} \right. \tag{4}$$

$$\left. -\frac{N-1}{N^2} \sum_{i=1}^N (\mathbf{u} \cdot \mathbf{x}^{(i)})^2 \sum_{i=1}^N (\mathbf{x}\mathbf{x}^T)^{(i)} - \frac{2N-2}{N^2} \left( \sum_{i=1}^N (\mathbf{u} \cdot \mathbf{x}^{(i)}) \mathbf{x}^{(i)} \right) \left( \sum_{i=1}^N (\mathbf{u} \cdot \mathbf{x}^{(i)}) \mathbf{x}^{(i)} \right)^T \right\}$$

Using (4) to estimate $\mathcal{H}_{\mathbf{u}} \kappa_4(\mathbf{u}^T \mathbf{x})$ from data when implementing Algorithm 1, the resulting quasi-orthogonalization algorithm runs in $O(Nd^3)$ time. Using (2) or (3) to estimate $\nabla_{\mathbf{u}} G(\mathbf{v}^T \mathbf{y})$ (with $G$ chosen to be $\kappa_3$ or $\kappa_4$ respectively) when implementing Algorithm 2 gives an update step that runs in $O(Nd)$ time. If $t$ bounds the number of iterations to convergence in Algorithm 2, then $O(Nd^2t)$ steps are required to recover all columns of $R$ once quasi-orthogonalization has been achieved.

## 6 Simulation Results

In Figure 1, we compare our algorithms to the baselines JADE [7] and versions of FastICA [10], using the code made available by the authors. Except for the choice of the contrast function for FastICA the baselines were run using default settings. All tests were done using artificially generated data. In implementing our algorithms (available at [19]), we opted to enforce orthogonality during the update step of Algorithm 2 with previously found columns of $R$. In Figure 1, comparison on five distributions indicates that each of the independent coordinates was generated from a distinct distribution among the Laplace distribution, the Bernoulli distribution with parameter 0.5, the t-distribution with 5 degrees of freedom, the exponential distribution, and the continuous uniform distribution. Most of these distributions are symmetric, making GI-$\kappa_3$ inadmissible.

When generating data for the ICA algorithm, we generate a random mixing matrix $A$ with condition number 10 (minimum singular value 1 and maximum singular value 10), and intermediate singular values chosen uniformly at random. The noise magnitude indicates the strength of an additive white Gaussian noise. We define 100% noise magnitude to mean variance 10, with 25% noise and 50% noise indicating variances 2.5 and 5 respectively. Performance was measured using the Amari Index introduced in [1]. Let $\hat{B}$ denote the approximate demixing matrix returned by an ICA algorithm, and let $M = \hat{B}A$. Then, the Amari index is given by: $E := \sum_{i=1}^n \sum_{j=1}^n \left( \frac{|m_{ij}|}{\max_k |m_{ik}|} - 1 \right) + \sum_{j=1}^n \sum_{i=1}^n \left( \frac{|m_{ij}|}{\max_k |m_{kj}|} - 1 \right)$. The Amari index takes on values between 0 and the dimensionality $d$. It can be roughly viewed as the distance of $M$ from the nearest scaled permutation matrix $PD$ (where $P$ is a permutation matrix and $D$ is a diagonal matrix).

From the noiseles data, we see that quasi-orthogonalization requires more data than whitening in order to provide accurate results. Once sufficient data is provided, all fourth order methods (GI-$\kappa_4$, JADE, and $\kappa_4$-FastICA) perform comparably. The difference between GI-$\kappa_4$ and $\kappa_4$-FastICA is not

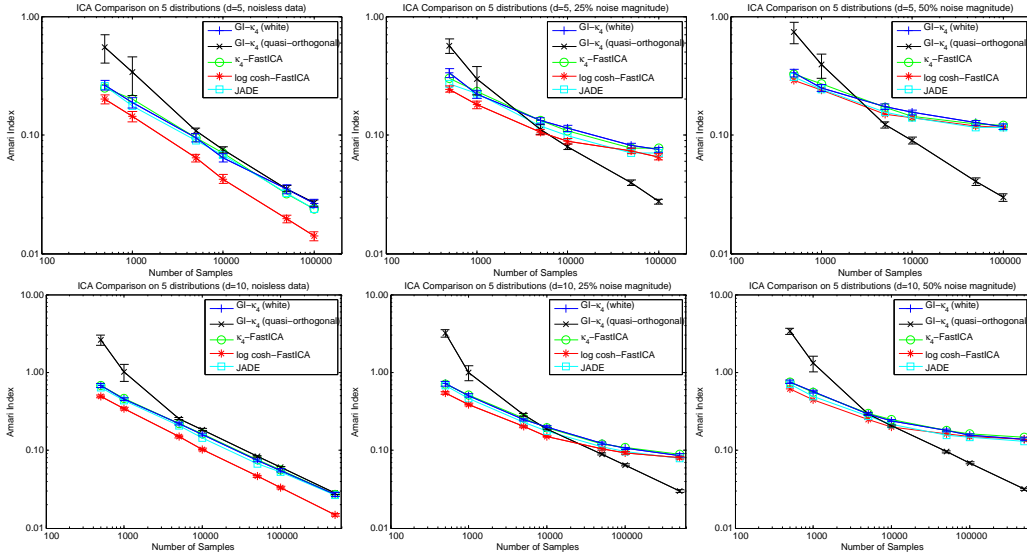

Figure 1: Comparison of ICA algorithms under various levels of noise. White and quasi-orthogonal refer to the choice of the first step of ICA. All baseline algorithms use whitening. Reported Amari indices denote the mean Amari index over 50 runs on different draws of both $A$ and the data. $d$ gives the data dimensionality, with two copies of each distribution used when $d = 10$.

statistically significant over 50 runs with 100 000 samples. We note that GI-$\kappa_4$ under whitening and $\kappa_4$-FastICA have the same update step (up to a slightly different choice of estimators), with GI-$\kappa_4$ differing to allow for quasi-orthogonalization. Where provided, the error bars give a $2\sigma$ confidence interval on the mean Amari index. In all cases, error bars for our algorithms are provided, and error bars for the baseline algorithms are provided when they do not hinder readability.

It is clear that all algorithms degrade with the addition of Gaussian noise. However, GI-$\kappa_4$ under quasi-orthogonalization degrades far less when given sufficient samples. For this reason, the quasi-orthogonalized GI-$\kappa_4$ outperforms all other algorithms (given sufficient samples) including the $\log \cosh$-FastICA, which performs best in the noiseless case. Contrasting the performance of GI-$\kappa_4$ under whitening with itself under quasi-orthogonalization, it is clear that quasi-orthogonalization is necessary to be robust to Gaussian noise.

Run times were indeed reasonably fast. For 100 000 samples on the varied distributions ($d = 5$) with 50% Gaussian noise magnitude, GI-$\kappa_4$ (including the orthogonalization step) had an average running time[2] of 0.19 seconds using PCA whitening, and 0.23 seconds under quasi-orthogonalization. The corresponding average number of iterations to convergence per independent component (at 0.0001 error) were 4.16 and 4.08. In the following table, we report the mean number of steps to convergence (per independent component) over the 50 runs for the 50% noise distribution ($d = 5$), and note that once sufficiently many samples were taken, the number of steps to convergence becomes remarkably small.

| Number of data pts | 500 | 1000 | 5000 | 10000 | 50000 | 100000 |
|---|---|---|---|---|---|---|
| whitening+GI-$\kappa_4$: mean num steps | 11.76 | 5.92 | 4.99 | 4.59 | 4.35 | 4.16 |
| quasi-orth.+GI-$\kappa_4$: mean num steps | 213.92 | 65.95 | 4.48 | 4.36 | 4.06 | 4.08 |

# 7 Acknowledgments

This work was supported by NSF grant IIS 1117707.

## Footnotes

[1]This process of orthogonalizing the latent signals was called *quasi-whitening* in [2] and later in [3]. However, this conflicts with the definition of quasi-whitening given in [12] which requires the latent signals to be whitened. To avoid the confusion we will use the term *quasi-orthogonalization* for the process of orthogonalizing the latent signals.

[2] Using a standard desktop with an i7-2600 3.4 GHz CPU and 16 GB RAM.

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
