[Supplementary Material · main_long_camera.pdf]

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

 [9] extends the Bayesian techniques to be robust to sparse noise When there are sufficiently more dimensions than underlying source signals, and when the additive noise in ICA does not include non-white Gaussian noise, these techniques can be viewed as alternatives for step 1 of ICA.

## 2   Using Cumulants to Orthogonalize the Independent Components

**Properties of Cumulants:** Cumulants are similar to moments and can be expressed in terms of certain polynomials of the moments. However, cumulants have additional properties which allow independent random variables to be algebraically separated. We will be interested in the fourth order multi-variate cumulants, and univariate cumulants of arbitrary order. Denote by $Q_{\mathbf{x}}$ the fourth order cumulant tensor for the random vector $\mathbf{x}$. So, $(Q_{\mathbf{x}})_{ijkl}$ is the cross-cumulant between the random variables $x_i, x_j, x_k$, and $x_l$, which we alternatively denote as $\mathrm{Cum}(x_i, x_j, x_k, x_l)$. Cumulant tensors

are symmetric, i.e. $(Q_{\mathbf{x}})_{ijkl}$ is invariant under permutations of indices. Multivariate cumulants have the following properties (written in the case of fourth order cumulants):

1. (Multilinearity) If $\mathbf{x}$ is a random vector, and $\alpha$ is a scalar, then

$$\mathrm{Cum}(\alpha x_i, x_j, x_k, x_l) = \alpha \, \mathrm{Cum}(x_i, x_j, x_k, x_l) \,.$$

If $y$ is a random variable, then

$$\mathrm{Cum}(x_i + y, x_j, x_k, x_l) = \mathrm{Cum}(x_i, x_j, x_k, x_l) + \mathrm{Cum}(y, x_j, x_k, x_l) \,.$$

2. (Independence) If $x_i$ and $x_j$ are independent random variables, then

$$\mathrm{Cum}(x_i, x_j, x_k, x_l) = 0 \,.$$

When $\mathbf{x}$ and $\mathbf{y}$ are independent, $Q_{\mathbf{x}+\mathbf{y}} = Q_{\mathbf{x}} + Q_{\mathbf{y}}$.

3. (Vanishing Gaussian) Cumulants of order 3 and above are zero for Gaussian random variables.

The first order cumulant is the mean, and the second order multivariate cumulant is the covariance matrix. We will denote by $\kappa_r(x)$ the order-$r$ univariate cumulant, which is equivalent to the cross-cumulant of $x$ with itself $r$ times: $\kappa_r(x) := \mathrm{Cum}(x, x, \ldots, x)$ (where $x$ appears $r$ times). In the univariate case, these properties become:

1. (Additivity) If $x$ and $y$ are independent random variables, then $\kappa_r(x+y) = \kappa_r(x) + \kappa_r(y)$.

2. (Homogeneity) If $\alpha$ is a scalar and $x$ a random variable, then $\kappa_r(\alpha x) = \alpha^r \kappa_r(x)$.

3. (Vanishing Gaussians) The first-order mean and second-order variance are the only non-zero cumulants of Gaussian random variables.

**Quasi-Orthogonalization Using Cumulant Tensors.** Recalling our original notation, $\mathbf{x} = A\mathbf{s} + \mathbf{b} + \boldsymbol{\eta}$ gives the generative ICA model. We define an operation of fourth-order tensors on matrices: For $Q \in \mathbb{R}^{d \times d \times d \times d}$ and $M \in \mathbb{R}^{d \times d}$, $Q(M)$ is the matrix such that

$$Q(M)_{ij} := \sum_{k=1}^{d} \sum_{l=1}^{d} Q_{ijkl} m_{lk} \,. \tag{1}$$

We can use this operation to orthogonalize the latent random signals.

**Definition 2.1.** A matrix $W$ is called a *quasi-orthogonalization* matrix if there exists an orthogonal matrix $R$ and a nonsingular diagonal matrix $D$ such that $WA = RD$.

We will need the following results from [3]. Here we use $A_q$ to denote the $q^{\text{th}}$ column of $A$.

**Lemma 2.2.** *Let $M \in \mathbb{R}^{d \times d}$ be an arbitrary matrix. Then, $Q_{\mathbf{x}}(M) = ADA^T$ where $D$ is a diagonal matrix with entries $d_{qq} = \kappa_4(s_q)A_q^T M A_q$.*

**Theorem 2.3.** *Suppose that each component of $\mathbf{s}$ has non-zero fourth cumulant. Let $M = Q_{\mathbf{x}}(I)$, and let $C = Q_{\mathbf{x}}(M^{-1})$. Then $C = ADA^T$ where $D$ is a diagonal matrix with entries $d_{qq} = 1/\|A_q\|_2^2$. In particular, $C$ is positive definite, and for any factorization $BB^T$ of $C$, $B^{-1}$ is a quasi-orthogonalization matrix.*

It is worth noting that if each component of $\mathbf{s}$ has fourth cumulant of the same sign, then $M = Q_{\mathbf{x}}(I)$ will be either positive or negative definite, and (negating $M$ if it is negative-definite) a factorization of $M$ could be used instead for quasi-orthogonalization. However, the inversion trick in Theorem 2.3 is necessary to cancel out the fourth-cumulant signs in the general case.

## 3 Quasi-Orthogonalization using Cumulant Hessians

We have seen in Theorem 2.3 a tensor-based method which can be used to quasi-orthogonalize observed data. However, this method naïvely requires the estimation of $O(d^4)$ terms from data. There is a connection between the cumulant Hessian-based techniques used in ICA [2, 11] and the tensor-based technique for quasi-orthogonalization described in Theorem 2.3 that allows the tensor-method to be rewritten using a series of Hessian operations. We make this connection precise

---

**Algorithm 1** Hessian-based algorithm to generate a quasi-orthogonalization matrix.

---

1: **function** FINDQUASIORTHOGONALIZATIONMATRIX($\mathbf{x}$)
2:     Let $M = \frac{1}{12} \sum_{i=1}^{d} \mathcal{H}_{\mathbf{u}} \kappa_4(\mathbf{u}^T \mathbf{x})|_{\mathbf{u}=\mathbf{e}_i}$. See Equation (6) for the estimator.
3:     Let $U \Lambda U^T$ give the eigendecomposition of $M^{-1}$
4:     Let $C = \sum_{i=1}^{d} \lambda_i \mathcal{H}_{\mathbf{u}} \kappa_4(\mathbf{u}^T \mathbf{x})|_{\mathbf{u}=U_i}$. See Equation (6) for the estimator.
5:     Factorize $C$ as $BB^T$.
6:     return $B^{-1}$
7: **end function**

---

below. The Hessian version requires only $O(d^3)$ terms to be estimated from data and simplifies the computation to consist of matrix and vector operations.

Let $\mathcal{H}_{\mathbf{u}}$ denote the Hessian operator with respect to a vector $\mathbf{u} \in \mathbb{R}^d$. The following lemma connects Hessian methods with our tensor-matrix operation (a special case is discussed in [2, Section 2.1]).

**Lemma 3.1.** $\mathcal{H}_{\mathbf{u}}(\kappa_4(\mathbf{u}^T \mathbf{x})) = ADA^T$ where $d_{qq} = 12(\mathbf{u}^T A_q)^2 \kappa_4(s_q)$.

*Proof.* By the properties of cumulants: $\kappa_4(\mathbf{u}^T A \mathbf{s}) = \sum_q (\mathbf{u}^T A_q)^4 \kappa_4(s_q)$. Taking derivatives yields

$$\partial_{u_i u_j} \kappa_4(\mathbf{u}^T A \mathbf{s}) = 12 \sum_q (\mathbf{u}^T A_q)^2 \kappa_4(s_q) a_{iq} a_{jq} .$$

In matrix form, this reads $\mathcal{H}_{\mathbf{u}} f(\mathbf{u}) = ADA^T$ where $D$ is a diagonal matrix with entries $d_{qq} = 12 \kappa_4(s_q)(\mathbf{u}^T A_q)^2$. $\qquad \square$

In Lemma 3.1, the diagonal entries can be rewritten as $d_{qq} = 12 \kappa_4(s_q)(A_q^T (\mathbf{u}\mathbf{u}^T) A_q)$. By comparing with Lemma 2.2, we see that applying $Q_{\mathbf{x}}$ against a symmetric, rank one matrix $\mathbf{u}\mathbf{u}^T$ can be rewritten in terms of the Hessian operations:

$$Q_{\mathbf{x}}(\mathbf{u}\mathbf{u}^T) = \frac{1}{12} \mathcal{H}_{\mathbf{u}}(\kappa_4(\mathbf{u}^T \mathbf{x})) .$$

This formula extends to arbitrary symmetric matrices by the following Lemma.

**Lemma 3.2.** *Let $M$ be a symmetric matrix with eigen decomposition $U \Lambda U^T$ such that $U = (\mathbf{u}_1, \mathbf{u}_2, \ldots, \mathbf{u}_d)$ and $\Lambda = \mathrm{diag}(\lambda_1, \lambda_2, \ldots, \lambda_d)$. Then,*

$$Q_{\mathbf{x}}(M) = \frac{1}{12} \sum_{i=1}^{d} \lambda_i \mathcal{H}_{\mathbf{u}_i} \kappa_4(\mathbf{u}_i^T \mathbf{x}) .$$

*Proof.* Use the linearity of summations and (1) to see:

$$Q_{\mathbf{x}}(M) = Q_{\mathbf{x}} \left( \sum_{i=1}^{d} \lambda_i \mathbf{u}_i \mathbf{u}_i^T \right) = \sum_{i=1}^{d} \lambda_i Q_{\mathbf{x}}(\mathbf{u}_i \mathbf{u}_i^T) = \frac{1}{12} \sum_i \lambda_i \mathcal{H}_{\mathbf{u}_i}(\mathbf{u}_i \mathbf{u}_i^T) . \quad \square$$

The matrices $I$ and $M^{-1}$ in Theorem 2.3 are symmetric. As such, the tensor-based method for quasi-orthogonalization can be rewritten using Hessian operations. This is done in Algorithm 1.

## 4   Gradient Iteration ICA

In the preceding sections, we discussed techniques to quasi-orthogonalize data. For this section, we will assume that quasi-orthogonalization is accomplished, and discuss deflationary approaches that can quickly recover the directions of the independent components. Let $W$ be a quasi-orthogonalization matrix. Then, define $\mathbf{y} := W\mathbf{x} = WA\mathbf{s} + W\boldsymbol{\eta}$. Note that since $\boldsymbol{\eta}$ is Gaussian noise, so is $W\boldsymbol{\eta}$. There exists a rotation matrix $R$ and a diagonal matrix $D$ such that $WA = RD$. Let $\tilde{\mathbf{s}} := D\mathbf{s}$. The coordinates of $\tilde{\mathbf{s}}$ are still independent random variables. Gaussian noise makes recovering the scaling matrix $D$ impossible. We aim to recover the rotation matrix $R$.

To see why recovery of $D$ is impossible, we note that a white Gaussian random variable $\boldsymbol{\eta}_1$ has independent components. It is impossible to distinguish between the case where $\boldsymbol{\eta}_1$ is part of the signal, i.e. $WA(\mathbf{s} + \boldsymbol{\eta}_1) + W\boldsymbol{\eta}$, and the case where $A\boldsymbol{\eta}_1$ is part of the additive Gaussian noise, i.e. $WA\mathbf{s} + W(A\boldsymbol{\eta}_1 + \boldsymbol{\eta})$, when $\mathbf{s}$, $\boldsymbol{\eta}_1$, and $\boldsymbol{\eta}$ are drawn independently. In the noise-free ICA setting, the latent signal is typically assumed to have identity covariance, placing the scaling information in the columns of $A$. The presence of additive Gaussian noise makes recovery of the scaling information impossible since the latent signals become ill-defined. Following the idea popularized in FastICA, we will discuss a deflationary technique to recover the columns of $R$ one at a time.

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

*Proof of Theorem 4.2.* Since $G$ is $0$ for Gaussian random variables, and since the noise separates by additivity, we can drop $W\boldsymbol{\eta}$ from all calculations. For simplicity, we will consider the change of coordinates $\mathbf{u} := R^{-1}\mathbf{v}$. As a special case of Lemma 4.1 where $R$ is the identity, we get:

$$\nabla_{\mathbf{u}} G(\mathbf{u} \cdot \tilde{\mathbf{s}}) = r \sum_{i=1}^{d} u_i^{r-1} G(\tilde{s}_i) \mathbf{e}_i . \tag{2}$$

**Claim 4.2.1.** $\nabla_{\mathbf{u}} G(\mathbf{u} \cdot \tilde{\mathbf{s}}) = R^{-1} \nabla_{\mathbf{v}} G(\mathbf{v} \cdot \mathbf{y})$.

*Proof of Claim.* Using Lemma 4.1:

$$\nabla_{\mathbf{v}} G(\mathbf{v} \cdot \mathbf{y}) = r \sum_{i=1}^{n} (\mathbf{v} \cdot R_i)^{r-1} G(\tilde{s}_i) R_i$$

$$= r \sum_{i=1}^{n} (R^{-1}\mathbf{v})_i^{r-1} G(\tilde{s}_i) R \mathbf{e}_i$$

$$= R \left( r \sum_{i=1}^{n} (u_i)^{r-1} G(\tilde{s}_i) \mathbf{e}_i \right) = R \nabla_{\mathbf{u}} G(\mathbf{u} \cdot \tilde{\mathbf{s}}) .$$

Multiplying both sides by $R^{-1}$ gives the Claim. ∎

$R^{-1}$ is the change of coordinates from the observed coordinate system to the coordinate system of the independent components. Hence, the update rule $\mathbf{v} \leftarrow \nabla_{\mathbf{v}} G(\mathbf{v} \cdot \mathbf{y})$ in the observed coordinate system has the same effect as the update rule $\mathbf{u} \leftarrow \nabla_{\mathbf{u}} G(\mathbf{u} \cdot \tilde{\mathbf{s}})$ in the coordinates of $\tilde{s}$. $\mathbf{v}$ converges to a column of $R$ (up to sign) precisely if $\mathbf{u}$ converges to some canonical vector $\pm \mathbf{e}_j$ (up to sign). It suffices to consider what happens in the coordinate system of $\tilde{s}$.

Without loss of generality, suppose that the coordinates of $\mathbf{u}$ are ordered such that:

$$|u_1^{r-2} G(\tilde{s}_1)| > |u_2^{r-2} G(\tilde{s}_2)| \geq \cdots \geq |u_d^{r-2} G(\tilde{s}_d)| \tag{3}$$

Since there exists $i$ such that $G(s_i) \neq 0$, and since $\mathbf{u}$ is drawn uniformly at random from the unit sphere along with $\mathbf{v}$, such an ordering can be achieved almost surely by reindexing $\mathbf{u}$. If any $u_i^{r-2} G(\tilde{s}_i) = 0$, the resulting value of $u_i$ after any update will always be $0$, ensuring that $\mathbf{u}$ does not converge to $\mathbf{e}_i$.

Let $m$ denote the largest index $i$ such that $|u_i^{r-2} G(\tilde{s}_i)| > 0$. If $m = 1$, then convergence will be achieved after one update, leaving nothing to prove. So, assume $m > 1$. Let $\mathbf{u}(k)$ denote $\mathbf{u}$ after the $k^{\text{th}}$ update, with $\mathbf{u}(0)$ denoting the input value of $\mathbf{u}$. Define for $i, j \in \{1, 2, \ldots, m\}$:

$$\rho(i, j; k) := \left| \frac{u_i(k)}{u_j(k)} \right|$$

$$g_{ij} := \left| \frac{G(\tilde{s}_i)}{G(\tilde{s}_j)} \right|$$

$$c_{ij} := \left| \frac{u_i(0)^{r-2} G(s_i)}{u_j(0)^{r-2} G(s_j)} \right| = \rho(i, j; 0)^{r-2} g_{ij}$$

where by assumption, $c_{1j} > 1$ for any $j > 1$. We will see that $\rho(i, j; k)$ has order $r - 1$ growth with respect to $k$:

$$\rho(i, j; k) = \left| \frac{u_i(k)}{u_j(k)} \right| = \left| \frac{u_i(k-1)^{r-1} G(\tilde{s}_i)}{u_j(k-1)^{r-1} G(\tilde{s}_j)} \right|$$

$$= \cdots = \rho(i, j; 0)^{(r-1)^k} g_{ij}^{\sum_{m=0}^{k-1} (r-1)^m}$$

$$= \left( c_{ij} g_{ij}^{-1} \right)^{[1/(r-2)](r-1)^k} g_{ij}^{\sum_{m=0}^{k-1} (r-1)^m}$$

**Algorithm 2** A fast algorithm to recover a single column of $R$ when $\mathbf{v}$ is drawn generically from the unit sphere. Equations (4) and (5) provide $k$-statistic based estimates of $\nabla_{\mathbf{v}}\kappa_3$ and $\nabla_{\mathbf{v}}\kappa_4$, which can be used as practical choices of $\nabla_{\mathbf{v}}G$ on real data.

---

1: **function** GI-ICA($\mathbf{v}, \mathbf{y}$)
2:    **repeat**
3:       $\mathbf{v} \leftarrow \nabla_{\mathbf{v}}G(\mathbf{v}^T\mathbf{y})$
4:       $\mathbf{v} \leftarrow \mathbf{v}/\|\mathbf{v}\|_2$
5:    **until** Convergence **return** $\mathbf{v}$
6: **end function**

---

where $\frac{1}{r-2}(r-1)^k = \sum_{m=0}^{k-1}(r-1)^m + \frac{1}{r-2}$ since $r-2$ can be written as $(r-1)-1$. We get:

$$\rho(i,j;k) = c_{ij}^{[1/(r-2)](r-1)^k} g_{ij}^{-1/(r-2)}$$

In particular,

$$\rho(1,j;k) = c_{1j}^{[1/(r-2)](r-1)^k} g_{1j}^{-1/(r-2)} \geq c_{1j}^{(r-1)^{k-1}} g_{1j}^{-1/(r-2)}.$$

Since $c_{1j} > 1$, it is clear that $\rho(1,j;k) \to \infty$ as $k \to \infty$ for any $j \in \{2,3,\ldots,m\}$. In particular, $\mathbf{u}(k) \to \mathbf{e}_1$ (up to sign) as $k \to \infty$. What remains to be seen is the order of convergence. Assume without loss of generality that $\mathbf{u}(k) \to \mathbf{e}_1$. There exists $K \in \mathbb{N}$ such that $k > K$ implies $\angle(\mathbf{e}_1, \mathbf{u}(k)) < \frac{\pi}{4}$. Assume $k > K$. Let $\theta = \angle(\mathbf{e}_1, \mathbf{u}(k+1))$. Since $\sin(\theta) > 0$ and $\cos(\theta) > 0$, it follows that $\sin(\theta) + \cos(\theta) \geq \sin^2(\theta) + \cos^2(\theta) = 1 \Rightarrow \sin(\theta) \geq 1 - \cos(\theta)$. Noting that $\sin(\theta) = \|\sum_{i=2}^m u_i(k+1)\mathbf{e}_i\|_2$ and $1 - \cos(\theta) = 1 - u_1(k+1)$, we get that $1 - u_1(k+1) \leq \|\sum_{i=2}^m u_i(k+1)\mathbf{e}_i\|_2 \leq \sum_{i=2}^m |u_i(k+1)|$. It follows:

$$\|\mathbf{e}_1 - \mathbf{u}(k+1)\|_2 \leq 1 - u_1(k+1) + \sum_{i=2}^m |u_i(k+1)| \leq 2\sum_{i=2}^m |u_i(k+1)| \leq 2\sum_{i=2}^m \rho(i,1;k+1)$$

Letting $h = \arg\max_{i=2}^m \rho(i,1;k+1)$, and letting $g_1 = \max_{i=2}^m g_{1i}$, we get:

$$\|\mathbf{e}_1 - \mathbf{u}(k+1)\|_2 \leq 2m\rho(h,1;k+1) = 2m\rho(h,1;k)^{r-1}g_{1h}$$
$$< 2m\cos^{-(r-1)}(\pi/4)\|\mathbf{e}_1 - \mathbf{u}(k)\|_2^{r-1}g_1$$

by noting that $u_h(k) \leq \|\mathbf{e}_1 - \mathbf{u}(k)\|_2$ and $u_1(k) > \cos(\pi/4)$. It follows:

$$\frac{\|\mathbf{e}_1 - \mathbf{u}(k+1)\|_2}{\|\mathbf{e}_1 - \mathbf{u}(k)\|_2^{(r-1)}} < 2m\cos^{-(r-1)}(\pi/4)g_1$$

which is a constant, giving order $r-1$ convergence.     $\square$

**Recovering all Independent Components.** As a Corollary to Theorem 4.2 we get:

**Corollary 4.3.** *Suppose $R_1, R_2, \ldots, R_k$ are known for some $k < d$. Suppose there exists $i > k$ such that $G(s_i) \neq 0$. If $\mathbf{v}$ is drawn uniformly at random from $S^{d-1} \cap \text{span}(R_1, \ldots, R_k)^\perp$ where $S^{d-1}$ denotes the unit sphere in $\mathbb{R}^d$, then Algorithm 2 with input $\mathbf{v}$ converges to a new column of $R$ almost surely.*

Since the indexing of $R$ is arbitrary, Corollary 4.3 gives a solution to noisy ICA, in Algorithm 3. In practice (not required by the theory), it may be better to enforce orthogonality between the columns of $R$, by orthogonalizing $\mathbf{v}$ against previously found columns of $R$ at the end of each step in Algorithm 2. We expect the fourth or third cumulant function will typically be chosen for $G$.

## 5   Time Complexity Analysis and Estimation of Cumulants

To implement Algorithms 1 and 2 requires the estimation of functions from data. We will limit our discussion to estimation of the third and fourth cumulants, as lower order cumulants are more statistically stable to estimate than higher order cumulants. $\kappa_3$ is useful in Algorithm 2 for non-symmetric distributions. However, since $\kappa_3(s_i) = 0$ whenever $s_i$ is a symmetric distribution, it is plausible that $\kappa_3$ would not recover all columns of $R$. When $\mathbf{s}$ is suspected of being symmetric, it

---

**Algorithm 3** Algorithm for ICA in the presence of Gaussian noise. $\tilde{A}$ recovers $A$ up to column order and scaling. $R^T W$ is the demixing matrix for the observed random vector $\mathbf{x}$.

For simplicity of the theoretical arguments, these algorithms have been discussed as if we know the distributions of the observed random variables and can make all required computations exactly. So, matrix multiplication against a random vector should be read as defining a new distribution. In practice, the random vectors would be replaced by data matrices, and function computations would be approximated as described in Section 5.

---

    **function** GAUSSIANROBUSTICA($G$, $\mathbf{x}$)
        $W = $ FINDQUASIORTHOGONALIZATIONMATRIX($\mathbf{x}$)
        $\mathbf{y} = W\mathbf{x}$
        $R\_columns = \emptyset$
        **for** $i = 1$ to $d$ **do**
            Draw $\mathbf{v}$ from $S^{d-1} \cap \mathrm{span}(R\_columns)^\perp$ uniformly at random.
            $R\_columns = R\_columns \cup \{\text{GI-ICA}(\mathbf{v}, \mathbf{y})\}$
        **end for**
        Construct a matrix $R$ using the elements of $R\_columns$ as columns.
        $\tilde{\mathbf{s}} = R^T\mathbf{y}$
        $\tilde{A} = (R^T W)^{-1}$
        **return** $\tilde{A}, \tilde{\mathbf{s}}$
    **end function**

---

is prudent to use $\kappa_4$ for $G$. Alternatively, one can fall back to $\kappa_4$ from $\kappa_3$ when $\kappa_3$ is detected to be near 0.

Denote by $z^{(1)}, z^{(2)}, \ldots, z^{(N)}$ the observed samples of a random variable $z$. Given a sample, each cumulant can be estimated in an unbiased fashion by its $k$-statistic. Denote by $k_r(z^{(i)})$ the $k$-statistic sample estimate of $\kappa_r(z)$. Letting $m_r(z^{(i)}) := \frac{1}{N}\sum_{i=1}^N (z^{(i)} - \bar{z})^r$ give the $r^{\text{th}}$ sample central moment, then

$$k_3(z^{(i)}) := \frac{N^2 m_3(z^{(i)})}{(N-1)(N-2)} \ , \ \ k_4(z^{(i)}) := N^2 \frac{(N+1)m_4(z^{(i)}) - 3(N-1)m_2(z^{(i)})^2}{(N-1)(N-2)(N-3)}$$

gives the third and fourth $k$-statistics [15]. However, we are interested in estimating the gradients (for Algorithm 2) and Hessians (for Algorithm 1) of the cumulants rather than the cumulants themselves. The following Lemma shows how to obtain unbiased estimates:

**Lemma 5.1.** *Let $\mathbf{z}$ be a $d$-dimensional random vector with finite moments up to order $r$. Let $\mathbf{z}^{(i)}$ be an iid sample of $\mathbf{z}$. Let $\alpha \in \mathbb{N}^d$ be a multi-index. Then $\partial_{\mathbf{u}}^\alpha k_r(\mathbf{u} \cdot \mathbf{z}^{(i)})$ is an unbiased estimate for $\partial_{\mathbf{u}}^\alpha \kappa_r(\mathbf{u} \cdot \mathbf{z})$.*

*Proof.* Let the derivatives below be with respect to $\mathbf{u}$. Since $k_r$ is unbiased, we have

$$\partial^\alpha \kappa_r(\mathbf{u} \cdot \mathbf{z}) = \partial^\alpha \mathbb{E}_{\mathbf{z}^{(i)}}[k_r(\mathbf{u} \cdot \mathbf{z}^{(i)})] = \mathbb{E}_{\mathbf{z}^{(i)}}[\partial^\alpha k_r(\mathbf{u} \cdot \mathbf{z}^{(i)})] \ .$$

To justify the second equality, we need to verify the conditions for differentiation under the integral sign. We have $(\mathbf{z}^{(i)}, \mathbf{u}) \mapsto k_r(\mathbf{u} \cdot \mathbf{z}^{(i)})$ is a multivariate polynomial. It is enough to expand it as a sum of monomials and justify the step for every monomial. Each monomial is of the form $f(\mathbf{u})g(\mathbf{z}^{(i)})$ where $g$ is of degree $r$. Thus, under our assumptions it is integrable over $\mathbf{z}^{(i)}$ for every fixed $\mathbf{u}$. Moreover,

$$\partial^\alpha \mathbb{E}_{\mathbf{z}^{(i)}}[f(\mathbf{u})g(\mathbf{z}^{(i)})] = \partial^\alpha f(\mathbf{u})\mathbb{E}_{\mathbf{z}^{(i)}}[g(\mathbf{z}^{(i)})] = \mathbb{E}_{\mathbf{z}^{(i)}}[\partial^\alpha f(\mathbf{u})g(\mathbf{z}^{(i)})]. \qquad \square$$

If we mean-subtract (via the sample mean) all observed random variables, then the resulting estimates are:

$$\nabla_{\mathbf{u}} k_3(\mathbf{u} \cdot \mathbf{y}) = \frac{3N^2 \sum_{i=1}^{N}((\mathbf{u} \cdot \mathbf{y}^{(i)})^2 \mathbf{y}^{(i)})}{N(N-1)(N-2)} \tag{4}$$

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

Run times were indeed reasonably fast. For 100 000 samples on the varied distributions ($d = 5$) with 50% Gaussian noise magnitude, GI-$\kappa_4$ (including the orthogonalization step) had an average running time[2] of 0.19 seconds using PCA whitening, and 0.23 seconds under quasi-orthogonalization. The corresponding average number of iterations to convergence per independent component (at 0.0001 error) were 4.16 and 4.08. We report the mean number of steps to convergence (per independent component) over the 50 runs in Figure 2 for the 50% noise distribution ($d = 5$), and note that once sufficiently many samples were taken, the number of steps to convergence becomes remarkably small. Finally, we note that for sufficiently skewed distributions, GI-$\kappa_3$ under whitening outperforms all baseline algorithms on noiseless data (see Appendix B). GI-$\kappa_3$ is consistent with quasi-orthogonalization under Gaussian noise.

## 7    Discussion and Future Work

We have demonstrated a practical method for performing ICA under Gaussian noise. The first key point is using a relaxation of whitening which is invariant to additive Gaussian noise. We provide the first practical implementation of quasi-orthogonalization, which like whitening orthogonalizes the

| Number of data pts | 500 | 1000 | 5000 | 10000 | 50000 | 100000 |
|---|---|---|---|---|---|---|
| whitening+GI-$\kappa_4$: mean num steps | 11.76 | 5.92 | 4.99 | 4.59 | 4.35 | 4.16 |
| quasi-orth.+GI-$\kappa_4$: mean num steps | 213.92 | 65.95 | 4.48 | 4.36 | 4.06 | 4.08 |

Figure 2: Mean number of steps to convergence for GI-$\kappa_4$. 50 runs on a suite of 5 distributions in 5 dimensions with $50\%$ Gaussian noise magnitude. Tests are on 5-dimensional data with a cap of $1\,000$ iterations.

latent independent signals. Quasi-orthogonalization is a relaxation since neither are the data scaled to have unit variance, nor are the columns of the quasi-orthogonalized mixing matrix $WA$ scaled to have unit norm. We provide a novel analysis of cumulant-based ICA algorithms which extends fast convergence rates seen in ICA to the quasi-orthogonal setting, resulting in a class of fast, Gaussian noise invariant algorithms.

We note that our update step extends naturally to include cumulant-like objects which are robust to sparse noise such as those presented in [20]. As noted earlier, their are techniques in the literature [5, 9] to perform PCA in the presence of sparse noise which could conceivably be used for the first step of ICA. It would be interesting to combine these techniques and experiment with ICA which is more robust to sparse noise.

## 8 Acknowledgments

This work was supported by NSF grant IIS 1117707.

## Footnotes

[1]This process of orthogonalizing the latent signals was called *quasi-whitening* in [2] and later in [3]. However, this conflicts with the definition of quasi-whitening given in [12] which requires the latent signals to be whitened. To avoid the confusion we will use the term *quasi-orthogonalization* for the process of orthogonalizing the latent signals.

[2] Using a standard desktop with an i7-2600 3.4 GHz CPU and 16 GB RAM.

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

## A   Welling's Robust Cumulants Revisited

We will give a brief overview of the construction of Welling's robust moments and cumulants from [20] as taylored to the univariate case in order to demonstrate that the resulting robust cumulants have versions of the additivity and homogeneity properties (for $\alpha$-robust cumulants of order $r > 2$) required for admissibility into Algorithm 2 in the Gaussian noise-free case. Before proceeding, it should be noted that these cumulants are designed to be robust to extreme outliers rather than Gaussian noise. As such, Welling's robust cumulants are not fully in-line with the quasi-orthogonalization technique of this paper, which is not designed with sparse noise in mind. However, there are versions of PCA designed to be robust against sparse noise which should be compatible with Welling's robust cumulants.

In this Appendix, we argue that Welling's robust cumulants are admissible in the Gaussian-noise free case to the ICA Algorithm 2 given a suitable whitening or quasi-orthogonalization step. While we outline many of the necessary definitions, we make no claim to completeness, and we refer the reader to sections 2 and 3 of [20] for additional explanation. Given quasi-orthogonal random vector $\mathbf{y} = R\tilde{\mathbf{s}}$, an $\alpha$-robust characteristic function can be defined for the random vector $\mathbf{y}$ as:

$$\psi_{\tilde{\mathbf{s}}}^{(\alpha)}(\mathbf{t}) := \mathbb{E}[\exp(i\alpha \mathbf{y}^T \mathbf{t})g(\alpha, \tilde{\mathbf{s}})]$$

where $g(\alpha, \tilde{\mathbf{s}})$ is defined via the multi-variate standard normal probability density function $\phi$ as $g(\alpha, \tilde{\mathbf{s}}) := \frac{\phi(\alpha\tilde{\mathbf{s}})}{\phi(\tilde{\mathbf{s}})} = \exp(-(1/2)(\alpha^2 - 1)\|\tilde{\mathbf{s}}\|_2^2)$. Then, the moments and cumulants are defined from the derivatives of $\psi_{\tilde{\mathbf{s}}}^{(\alpha)}(\mathbf{t})$ and the second characteristic function $\Psi_{\tilde{\mathbf{s}}}^{(\alpha)}(\mathbf{t}) := \log_e \psi_{\tilde{\mathbf{s}}}^{(\alpha)}(\mathbf{t})$ respectively in the standard fashion. Under the orthogonality constraints of ICA, and noting that $\|\cdot\|_2$ is invariant under rotations, then $g(\alpha, \mathbf{y}) = g(\alpha, \tilde{\mathbf{s}})$. We can write:

$$\psi_{\tilde{\mathbf{s}}}^{(\alpha)}(\mathbf{t}) = \mathbb{E}\left[\exp(i\alpha \mathbf{y}^T \mathbf{t})g(\alpha, \mathbf{y})\right]$$

which gives the form required for producing sample estimates.

When defining the robust moments (and cumulants) in the direction of unit vector $\mathbf{u}$, we use the following directional $\alpha$-robust first characteristic function:

$$\psi_{\tilde{\mathbf{s}},\mathbf{u}}^{(\alpha)}(t) := \mathbb{E}[\exp(i\alpha(\mathbf{u} \cdot \mathbf{y})t)g(\alpha, \tilde{\mathbf{s}})]$$

The robust moments in direction $\mathbf{u}$ take on the form:

$$\mu_{r,\tilde{\mathbf{s}},\mathbf{u}}^{(\alpha)}(\mathbf{y}) := \alpha^r \mathbb{E}[(\mathbf{u} \cdot \mathbf{y})^r g(\alpha, \tilde{\mathbf{s}})] = \alpha^r \mathbb{E}[(\mathbf{u} \cdot \mathbf{y})^r g(\alpha, \mathbf{y})] .$$

where the last equality lends itself to sampling estimates.

The robust cumulants $\kappa_{r,\tilde{\mathbf{s}},\mathbf{u}}^{(\alpha)}$ are related to the robust moments using the typical moment expansions, except that $\mu_{0,\mathbf{u}}^{(\alpha)}$ cannot be assumed 1, and $\mu_{1,\mathbf{u}}$ cannot be assumed 0 for centered data as is generally

done. We refer the reader to Appendix A of [20] for the expansions of the $\alpha$-robust cumulants in terms of $\alpha$-robust moments. The following Theorem gives a version of the additivity and homogeneity properties for robust cumulants. These versions are consistent with the proof of Lemma 4.1 and hence the proof of Theorem 4.2 in the noise-free case, giving that the $\alpha$-robust cumulants are admissible to Algorithm 2:

**Theorem A.1.** *Let $r > 2$. Then the $\alpha$-robust cumulant $\kappa_{r,\tilde{\mathbf{s}},\mathbf{u}}^{(\alpha)}$ is additive in that*

$$\kappa_{r,\tilde{\mathbf{s}},\mathbf{u}}^{(\alpha)}(\mathbf{y}) = \sum_{j=1}^{d} \kappa_{r,\tilde{s}_i}^{(\alpha)}((\mathbf{u} \cdot R_i)\tilde{s}_i) .$$

$\kappa_{r,\tilde{s}_i,\mathbf{u}}^{(\alpha)}$ *is homogeneous of order $r$ in that*

$$\kappa_{r,\tilde{s}_i}^{(\alpha)}((\mathbf{u} \cdot R_i)\tilde{s}_i) = (\mathbf{u} \cdot R_i)^r \kappa_{r,\tilde{s}_i}^{(\alpha)}(\tilde{s}_i) .$$

*Proof.* To see additivity, it suffices to show that the directional second characteristic function has the desired additivity property:

$$\begin{aligned}
\Psi_{\mathbf{u},\tilde{\mathbf{s}}}^{(\alpha)}(t) &= \log \mathbb{E}[\exp(i\alpha(\mathbf{u} \cdot \mathbf{y})t)g(\alpha, \tilde{\mathbf{s}})] \\
&= \log \mathbb{E}\left[ \exp\left( i\alpha\Big(\mathbf{u} \cdot \sum_{i=1}^{d} R_i\tilde{s}_i\Big)t \right) \exp\left( -\frac{1}{2}(\alpha^2 - 1)\|\tilde{\mathbf{s}}\|_2^2 \right) \right] \\
&= \log \prod_{j=1}^{d} \mathbb{E}[\exp\left( i\alpha(\mathbf{u} \cdot R_j\tilde{s}_j)t \right) g(\alpha, \tilde{s}_j)] \\
&= \sum_{j=1}^{d} \log \mathbb{E}[\exp(i\alpha((\mathbf{u} \cdot R_j)\tilde{s}_j)t)g(\alpha, \tilde{s}_j)] .
\end{aligned}$$

To see the homogeneity property, we note that the $r^{\text{th}}$ $\alpha$-robust cumulant has a moment expansion which consists of a sum of terms where each term has order $r$. Thus, it suffices to show that the $\alpha$-robust moments have the desired homogeneity property:

$$\mu_{r,\tilde{s}_i}((\mathbf{u} \cdot R_i)\tilde{s}_i) = \alpha^r \mathbb{E}[((\mathbf{u} \cdot R_i)\tilde{s}_i)^r g(\alpha, \tilde{s}_i)] = \alpha^r(\mathbf{u} \cdot R_i)^r \mathbb{E}[\tilde{s}_i^r g(\alpha, \tilde{s}_i)] = (\mathbf{u} \cdot R_i)^r \mu_{r,\tilde{s}_i}(\tilde{s}_i) .$$

$\square$

Note that we have not explored the properties of Welling's robust cumulants under additive Gaussian noise. While these robust cumulants are known to be zero for Gaussian random variables, we have only shown additivity for independent random variables which operate in orthogonal directions. As such, we do not claim that Welling's robust cumulants are admissible under heavy Gaussian noise like the traditional cumulants.

## B   Skew Data Plot

Comparison of ICA algorithms on skewed data. All latent random variables are drawn independently from the Exponential distributions, and mixed similarly to the other experiments from Section 6. Notably, GI-$\kappa_3$ (white) compares favorably against all baselines. We do not compare against the skew-based implementation of FastICA because the update steps happen to be the same due to the moment expansion of $\kappa_3$.