[Reviews · NeurIPS 2013]

Submitted by Assigned_Reviewer_1

This paper presents a fast ICA algorithm that works best under Gaussian noise. This is demonstrated with components simulated from different univariate distributions and variable Gaussian noise.

The writing is clear. The paper is incremental in the sense that it builds on ideas from (Belkin et. al, 2013) but focuses on speeding up and improving their cumulant-based approach.
This is achieved via
1) a Hessian expansion of the cumulant-tensor-based quasi-orthogonalization.
2) gradient-based iterations that preserve quasi-orthogonalization of the latent factors (noised case) as well as whitening in the noiseless case.

Significance:
In this manner the authors manage to deliver a noise-invariant estimator of the mixing matrix A and sources s, while improving significantly on the dimensionality complexity.

Other notes:
- As you point out, your method weaknesses come out (inferior, unstable and slower) in the small data regime (less than 1000), due to quasi-orthogonalization.
Summary: The authors present a new robust ICA cumulant-based approach. Results are significant under Gaussian noise-corrupted data and would be of interest to the ICA community.

I've read the author's rebuttal.

Submitted by Assigned_Reviewer_4

This paper proposes a cumulant based independent component analysis (ICA) algorithm for source separation in the presence of additive Gaussian noise. The algorithm is somewhat incremental building upon Refs [2] and [3], but appear technically correct with experimental results confirming the claims made. The algorithms used for benchmarking assume no additive noise but is like InfoMax often quite robust to addition of noise.

The ICA literature is huge and many algorithms exist for ICA with additive noise for example quite a lot Bayesian using deterministic approximative inference as well as MCMC. For algorithms in the former category from around year 2000 see for example http://www.mitpressjournals.org/doi/pdf/10.1162/089976601753196003, http://www.mitpressjournals.org/doi/pdf/10.1162/089976602317319009, http://citeseerx.ist.psu.edu/viewdoc/summary?doi=10.1.1.37.597 Another field where the studied model is used all time is in factor analysis with identifiable source models. It is therefore doubtful that the proposed algorithm will really fill a void for ICA in the presence of noise.
Summary: This paper proposes a cumulant based independent component analysis algorithm for source separation in the presence of additive Gaussian noise. The algorithm is somewhat incremental building upon Refs [2] and [3], but appear technically correct with experimental results confirming the claims made.

Submitted by Assigned_Reviewer_6

The authors propose new algorithms for determining independent components, when the data is corrupted by Gaussian noise. The results of alternative methods are affected by noise. The algorithm improves on the runtime of a method based on the 4th cumulant.

The problem is relevant. Quality of the paper is high. The approach seems novel and the contributions significant.

The experimental results are not totally convincing. In synthetic experiments most of the variants of ICA perform similarly. For a small number of samples, the quasi-orthogonalization based method performs significantly worse than all other methods. For a large number of samples, though, the method performs at least comparable, even in the noiseless setting. A benefit is demonstrated for a high Gaussian noise setting, if sufficient data is provided.

It would be interesting how the different ICA methods compare in various real world applications, where the noise distribution is unknown a priori.
Runtime experiments are provided only for the method at hand. To set these into context, the runtime of alternative methods should be included.
Summary: New methods for the relevant problem of recovering independent components from data corrupted by Gaussian noise are proposed. General quality of the paper is high, the approach seems novel and the contributions significant.
Author Feedback

Author rebuttal: We would like to thank the reviewers for their useful comments. We believe that our approach to noise-invariant ICA is different from existing methods in the extensive ICA literature. We hope that our method will be useful in practice.